# Newborn Screening Long Term Follow-Up in the Medical Home

**DOI:** 10.3390/ijns5030025

**Published:** 2019-07-25

**Authors:** Deborah Badawi, Katharine Bisordi, Marilyn J. Timmel, Scott Sorongon, Erin Strovel

**Affiliations:** 1Maryland Department of Health, Baltimore, MD 21201, USA; 2Department of Pediatrics, Division of Human Genetics, University of Maryland School of Medicine, Baltimore, MD 21201, USA; 3University of Maryland School of Medicine Clinical and Translational Research Informatics Center, Baltimore, MD 21201, USA

**Keywords:** newborn screening, genetics, long-term follow-up

## Abstract

This demonstration project explored the feasibility of utilizing data from pediatric primary care providers to evaluate the long-term outcomes of children with disorders identified by newborn screening (NBS). Compliance with national guidelines for care and the morbidity for this population was also examined. Primary care practices were recruited and patients with sickle cell disease or who were deaf/hard of hearing were given the opportunity to enroll in the study. Data were collected on the quality of the medical home with practice data compared to family responses. Clinical outcomes for each patient were assessed by review of medical records and patient surveys. These data sources were compared to determine accuracy of primary care data, morbidity, and receipt of preventive care. Electronic data sharing was explored through transmission of Clinical Document Architecture (CDA) files. Care coordination was a challenge, even in highly accredited medical homes. Providers did not have complete information regarding clinical outcomes and children were not consistently receiving recommended preventive care. Electronic data sharing with public health departments encountered interface challenges. Primary care providers in the USA should not currently be used as a sole source to evaluate long-term outcomes of children with disorders identified by NBS.

## 1. Introduction

The promise of newborn screening (NBS) maintains that infants identified with a condition at birth will have a better clinical outcome compared to those identified clinically after becoming symptomatic. This requires early diagnosis of the condition, as well as appropriate treatment and preventive care that is available and accessible, ideally coordinated through a medical home. Collaborative and coordinated care has been demonstrated to meet the triple aims of public health: improving health care outcomes, being cost-effective, and providing improved patient experience [1,2].

This project investigates the feasibility of collecting long-term follow-up (LTFU) data from primary care providers on children with special health care needs identified through NBS. The recommendations of the Health and Human Services Advisory Committee on Heritable Disorders in Newborns and Children (ACHDNC) [3] and the Medical Home Workgroup for the National Coordinating Center for the Regional Genetic and Newborn Screening Service Collaboratives [4] advise that outcomes of interest include:Receipt of coordinated care through a medical homeReceipt of preventive careClinical outcomesOpportunity to enroll in clinical research studies

NBS is often perceived as a state-based public health program consisting of newborn dried blood spot screening. In reality, NBS is a multicomponent system of education, screening, diagnosis, treatment, and long-term follow-up. State-based NBS programs (laboratory and follow-up) work successfully with birthing hospitals to provide blood spot and point of care screening tests to almost 100% of newborns born in the US. The NBS programs typically follow infants with positive screens until they receive a diagnostic evaluation and specialty referrals. However, limited information is available regarding long-term outcomes for these children with special health care needs.

## 2. Patients and Methods

Children identified through NBS with sickle cell disease (SCD), or who were deaf/hard of hearing (DHH) were the target populations of this project. These disorders were chosen for three reasons. First, there are national projects describing follow-up efforts for these disorders. Second, Maryland already has a framework in place for LTFU for these conditions. Third, collaboration exists among the NBS follow-up program, providers, and family advocates for these groups of children. The following outcomes were measured:How do family ratings of the medical home indicators compare to ratings by practice staff?What percent of participating children have a care coordination plan that is regularly updated?What are the clinical outcomes of participating children in terms of?
Receipt of care aligned with best practices for their disorder?Acute and chronic complications?What percent of participating families were offered and are participating in research studies?Do primary care practices have the information available to answer questions regarding long-term clinical outcomes for their patients? How can this information be efficiently shared with public health programs?

### 2.1. Medical Home Measurement

The Medical Home Index (MHI) is a validated self-assessment and classification tool designed to measure behaviors and processes of care within any office setting. The Medical Home Family Index (MHFI) is a companion tool which collects similar information from the perspective of individual families within a practice [5,6]. The MHI allows a practice to rate itself in 6 domains as a Level 1 through 4 medical home. Three of these domains are relevant to this project: organizational capacity for children with special health care needs (CSHCN) and their families, chronic condition management, and care coordination. A medical home rating level of 3 or 4 in all 3 domains was considered an affirmative answer to receipt of coordinated care in a medical home. A rating level of 1 or 2 in any or all of these domains was considered a negative answer.

The MHFI asks families whether certain medical home qualities are present Never, Sometimes, Often, or Always in their provider’s practice. These ratings were translated into scores of 1–4. The MHFI is not divided into separate domains, and consensus among study staff was used to determine which questions would comprise appropriate scales correlated with the 3 domains of interest. The MHI and MHFI were assessed separately.

The specific question of whether or not each participating child had a regularly updated care coordination plan was incorporated into the clinical outcomes checklist (see below).

### 2.2. Clinical Outcomes

The Longitudinal Pediatric Data Resource of the Newborn Screening Translational Research Network (NBSTRN) [7] and the work of the Follow-Up and Treatment Workgroup of the ACHDNC provided common elements desired for NBSLTFU. These include standards of care for a particular disorder when available, as well as indicators of well-being and potential complications. This framework, together with clinical guidelines from the National Heart, Lung and Blood Institute [8] for SCD, and the Joint Commission on Infant Hearing [9] for DHH children, were used to create a clinical outcomes checklist for each condition. These checklists were then reviewed by local SCD experts and Early Hearing Detection and Intervention stakeholders. The final drafts were sent to the Steering Committee of the New York Mid Atlantic Consortium for Genetic and Newborn Screening Services for input. The clinical outcomes checklists for SCD and DHH can be found in Appendix A and B, respectively.

For each patient participating in the study, a caregiver and a member of the medical home staff completed the clinical outcomes checklist. These were compared to assess their level of agreement and information available to each respondent.

### 2.3. Electronic Data Sharing

The final goal of this project was to determine the capacity for electronic data sharing between the primary care provider and the public health database. Maryland was collecting follow-up data using the state Health Information Exchange (HIE), the Chesapeake Region Information System for our Patients (CRISP), for the state newborn hearing screening program. For SCD, information was collected from providers via phone or fax and then entered into a state database. A dummy copy of the state sickle cell database was created and participant names entered. Project staff worked with CRISP and the MDH to explore messaging formats that would allow collection of NBSLTFU data efficiently and confidentially.

### 2.4. Participants

We solicited participation among primary care practices by direct invitation and outreach through the Maryland Chapter of the American Academy of Pediatrics. The goal was to recruit geographically and demographically diverse practice populations. Three such practices were recruited. The first was a private multisite practice in central Maryland. The second was a single location private practice in rural southern Maryland, and the third was an academic center outpatient practice located in the city of Baltimore. All three are National Center for Quality Assurance designated Patient-Centered Medical Homes, with a Level 3 accreditation. Participants were recruited via invitation letters coming directly from each practice. In addition, study staff directly contacted families from the participating sites who were already in the MDH sickle cell LTFU program. Fourteen total participants were recruited, 8 with SCD and 6 with DHH. Approval from the Maryland Department of Health and Mental Hygiene Institutional Review Board was obtained on August 24, 2013, Protocol #13-45.

## 3. Results

Aggregate data, as well as data collection instruments, can be accessed by contacting the corresponding author.

### 3.1. Medical Home

All three medical home practices completed the MHI, with at least 3 different staff members completing individual MHI surveys at each site. Staff completing the MHI included physicians, nurse care coordinators, and administrators. One parent or guardian of each participating patient also completed the MHFI.

Results are provided in Figure 1. Due to the small number of participants, trends are reported rather than levels of significance. Overall, families rated their medical home providers higher than staff rated their own practice. In contrast, care coordination was rated highest among practice staff and lowest by families.

Only half of enrolled children had an individual care plan in place. The proportion of children with a care plan in place was the same for private and academic center-based practices. Six of the 7 children who had a care plan had it updated periodically (5 regularly and 1 sometimes).

### 3.2. Clinical Outcomes

All children with SCD and 5 of the 6 DHH children were identified by newborn screening. One of the DHH children identified by newborn screening was diagnosed after 2 years of inconclusive diagnostic testing. The sixth child was identified upon follow-up screening due to having a neonatal risk factor.

Clinical outcomes were evaluated based on whether or not children received recommended standards of care, and specifically for children with SCD, the number and severity of complications. A clinical outcomes checklist was completed by each child’s primary care provider and caregiver. For each question regarding preventive care, the family was asked to indicate if the care was ever received and if it was received within the past year. Similarly, for complications of SCD, families were asked if each complication ever occurred and if it occurred within the past year. Electronic medical record (EMR) data on each child was reviewed to identify parallel information, specifically preventive care ever received and received in the past year, as well as complications listed in the child’s problem list, and those occurring in the past year based on visit documentation. The results were analyzed qualitatively to determine how many children received standards of care, and quantitatively to determine agreement between clinician and caregiver responses.

Upon review of standards of care for DHH children, we found that treatment with amplification was initiated as early as 6 months and as late as 8 years of age, although all were children diagnosed by 2 years of age. Two-thirds of the children were seen regularly by audiology during the first 2 years after receiving amplification. Genetic counseling was offered and completed by half of the families, whereas none were offered or participated in clinical research. All DHH children were offered and were receiving special education services.

Children with SCD only received some standards of care consistently. Annual influenza, PCV 13, and MCV4 vaccines, as well as penicillin where age appropriate, were consistently provided. Appropriate dosage increases starting at 3 years of age could not be confirmed (see Section 3.3. Electronic Data Sharing). Uncertainty existed among both parents and primary care providers regarding whether children had received the PPV23 vaccine. None received transcranial Doppler screening or hydroxyurea, yet all children saw a hematologist in the past year. Seven of the 8 mothers and 2 of the fathers knew their own sickle cell status prior to the child’s birth. Three mothers and 2 fathers were offered genetic counseling prenatally. Three mothers and 1 father were offered genetic counseling after the infant was diagnosed with SCD, and one set of these parents learned their sickle trait status. Half of the families (4) were asked to participate in a clinical research study and 3 accepted.

The occurrence and frequency of acute complications were evaluated for children with SCD. Six of 7 children experienced pain crises with the frequency varying from 1 to 13. One child had no data on this question. Five of the 8 children had emergency room visits in the past year, with the frequency varying from 1 to 10 visits. Half of the children had been hospitalized within the past year, with the frequency varying from 2 to 6 occurrences. Finally, 3 children had chest syndrome at some time in the past and 1 child had a stroke.

Significant knowledge gaps existed for both pediatricians and parents, with less than half of the clinical outcomes questions being answered by both parties. When parents and providers did answer the same question, their answers were in agreement 75% of the time. For children who are DHH, primary care providers are often not aware of the timing of amplification or audiology visits, if genetic counseling services offered or received, or if they were offered and accepted opportunities to participate in research studies. Only half of the providers were aware of any special education/early intervention services that were received. For children with SCD, families most often lacked information about specific vaccine status, with the exception of the flu vaccine. Pediatricians did not have information regarding whether genetic counseling was offered/accepted or whether research study participation was offered/accepted. They also had incomplete information regarding SCD-related complications, such as the number of ER visits and hospital admissions. Finally, pediatricians often did not have information about receipt of early intervention or special education services.

### 3.3. Electronic Data Sharing

Electronic data sharing between the medical home and public health database was piloted in this study. To achieve this, we used CRISP, which receives information from all hospitals as well as the majority of pharmacies, laboratories, and imaging centers in Maryland. In the two years preceding this project, CRISP sent alerts to those pediatricians signed up to receive them whenever a patient on their panel was seen in the emergency room or admitted to the hospital. This information was able to be entered into the patient’s record by the provider’s office, but not necessarily in a manner that could be easily identified and exported to an electronic report.

This project focused on developing electronic data sharing for children with SCD, given the newborn hearing screening program has an established data tracking system that is developing interfaces with primary care offices. The SCD follow-up program uses a database designed by program staff and receives information from primary care practices via fax or phone. CRISP and the project team worked together to identify shareable fields, in the hopes of creating a Clinical Document Architecture (CDA) file that could be sent from the provider electronic medical record (EMR) to the state database. This proved to be quite challenging, even when both private practices used the same EMR vendor and the University hospital-based practice used EPIC, a mainstream EMR. Practices using the same private vendor often have different software modification packages and levels of support. As a result, one practice was capable of exporting a CDA while the other was not, even though they used the same EMR vendor. Sending the CDA on a semi-annual or annual basis for each patient could not be automated without an additional cost for the private practice. In addition, although the private EMR incorporates standard fields, individual practices do not use these fields in the same way. For example, one practice identified subspecialty referrals by provider name, whereas the other practice identified them by specialty. Only the EPIC CDA could potentially be used to automatically send usable reports to the health department. The technical challenge we faced was ensuring that patients are appropriately matched, and we found this to be feasible.

## 4. Discussion

The findings of this project confirm that NBS is successful at identifying children with SCD and those who are DHH and getting them into care. A significant public health challenge is our ability to track the long-term outcomes of these children regarding receipt of recommended care and development of complications. On an individual basis, even this small demonstration project demonstrates the challenges of communication and care coordination for families and the medical home. These public health and individual clinical barriers are not unique to these diagnoses since the same systems of care exist for all children with conditions identified on NBS.

For children who are DHH, challenges remain in the timeliness of follow-up and the diagnostic odyssey to final diagnosis. In the small sample size of our study, one child underwent 2 years of repeat hearing tests before receiving a diagnosis. Best practices for follow-up have been developed and are available through the National Center for Hearing Assessment and Management [10].

For children with SCD, we found current guidelines are not being met. Children received vaccinations and prophylactic penicillin but did not receive hydroxyurea or transcranial Doppler (TCD) screening. Hematologists note that insurance coverage for hydroxyurea and access to appropriate facilities for TCD screening are barriers (private communication). Many quality improvement projects supported by HRSA’s Sickle Cell Treatment Demonstration project funding are working to identify tools to improve long term follow-up care.

Evaluation of the role of primary care providers in NBSLTFU revealed that providers have incomplete information regarding care and clinical outcomes for these children. Data is not always available from subspecialists and hospitals or it is difficult to access in scanned notes within the EMR. For example, if a child was fit for hearing aids or has an ER visit for a pain crisis, the primary care provider can be notified, but there is no consistent place to store this information for easy retrieval and transmission.

Providers who participate in research studies are typically very motivated to improve the quality of their practice and likely are higher performing than typical practices. Taking this into consideration, it is notable that practices thought they were doing better at care coordination than did their patients. In addition, only half of enrolled children have an individual care plan in place. Barriers to improved performance include staff time, knowledge of available resources, and communication among care team members.

Finally, electronic data sharing between private practices and public health is not feasible at this point in time. The Office of the National Coordinator for Health Information Technology (ONC), and its state and local partners continue to address data interface challenges. The lack of consistency in the infrastructure of individual practice EMRs do not allow standardized reports to be created and shared, and individual reports cannot be created for each provider. It is possible that large hospital-based practices are more appropriate targets for collecting information regarding NBSLTFU. These practices have both the advantage of using more limited numbers of EMRs and having subspecialty data available on children. However, in these larger systems of care, there are often numerous demands for modifications to the EMR and public health reporting on NBSLTFU may not be a priority for IT departments.

We propose technical solutions that are quite feasible. EMR vendors could develop standardized data fields based on national standards, so there would not be variation in the type of data entered in them. Moreover, EMR vendors could be given incentives to create standardized NBSLTFU data reports. Guidance regarding data elements to be collected could be provided by the ACHDNC in collaboration with other national organizations such as NBSTRN, NewSTEPs, and the Genetic Alliance. State health departments could then utilize these reports to track long-term outcomes for NBSLTFU.

This project was designed as a demonstration effort, and although limited by the number of cases, valuable information was obtained for further study. Potential solutions to the gaps identified exist. Quality improvement efforts are needed to target medical home capacity for NBSLTFU specifically and CSHCN in general. In order to fulfill the promise of NBS, it is not enough to identify and refer children. We must assure that they are able to receive evidence-based care throughout their lives so that they can reach the best outcome possible.

## 5. Conclusion

Newborn screening is considered a public health success because the majority of newborns born in the US receive screening for a recommended panel of treatable genetic conditions. The NBS community believes that morbidity and mortality for many disorders have been significantly reduced by NBS, but due to the lack of a nationwide system to collect health information on affected newborns identified by NBS, we are not able to quantify or describe this impact. This is a missed opportunity to tell the complete story of NBS successes.

## Figures and Tables

**Figure 1 IJNS-05-00025-f001:**
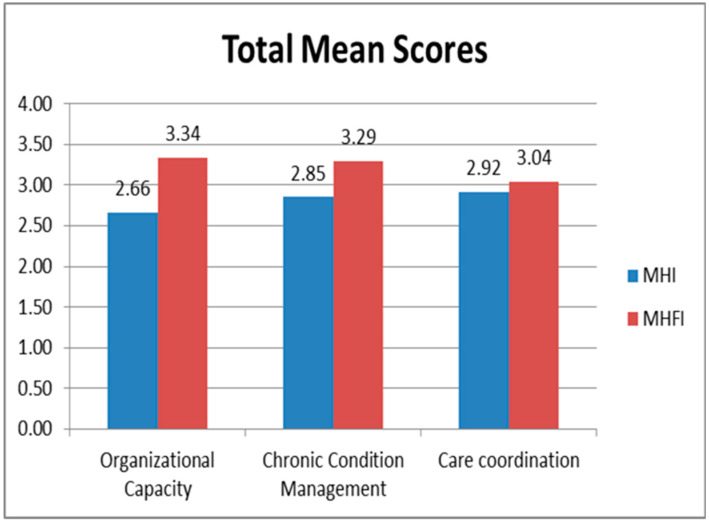
Medical home rating scales comparison of practice staff and family ratings on key medical home components.

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
