# Peer review of "Newborn Screening Long Term Follow-Up in the Medical Home"

_2409-515X, 2019, doi:10.3390/ijns5030025_

Reviewer 1 Report

This manuscript describes the perceptions of follow-up adequacy between the medical home and the families of babies identified with sickle cell disease or hearing deficit by newborn screening. 

It is unfortunate that so few responses were received from the families and the medical homes. However, it is evident from the responses that clinical follow-up of both sickle cell disease and hearing deficit is irregular at best and that the follow-up is deficient both in the understanding by the medical homes and the perception of the families about the adequacy of this understanding as well as the quality of the follow-up care. Thus, there is still a great deal of work needed in the medical follow-up of newborn screening identifications. My guess is that the results of this study in Maryland reflects the general status of newborn screening follow-up throughout the United States. 

The "Table 1" is a figure, not a table, and should be reclassified as Figure 1 in the legend and in the text. 

Author Response

Thank you for your thoughtful review.  We have relabeled Figure 1 appropriately in the revised manuscript.  

Reviewer 2 Report

The manuscript by Badawi, et al., begins to address pertinent questions regarding long term follow-up for children identified by NBS. However, the paper falls short in providing new information or techniques by which to collect and utilize this information and is hampered by very low sample sizes, especially for two relatively common conditions.

The manuscript could be strengthened by a more robust discussion and definition of long term follow-up, both in public health and clinically.

Additionally, it would be helpful to indicate demographics of the participating families to note whether there may be any trends in scoring and whether they may be confounding variables leading to lack of access to treatment, etc.

There are areas of clarity needed within the clinical outcomes results as well. For example, what timeline were the frequency of pain crises measured; per year? per month? Were all ER visits/hospital admissions secondary to the studied condition or for other reasons?

Author Response

Thank you very much for your thoughtful review and comments.  

Point 1: The manuscript could be strengthened by a more robust discussion and definition of long term follow-up, both in public health and clinically.

Point 1 response: There have been edits made to the discussion in the manuscript to address the need to outline both public health and clinical concerns regarding long term follow up of NBS.

Point 2: Additionally, it would be helpful to indicate demographics of the participating families to note whether there may be any trends in scoring and whether they may be confounding variables leading to lack of access to treatment, etc.

Point 2 response: Demographic data was not analyzed due to the small sample size.  All of the children with SCD were African American, and the DHH group was predominantly white.  

Point 3: There are areas of clarity needed within the clinical outcomes results as well. For example, what timeline were the frequency of pain crises measured; per year? per month? Were all ER visits/hospital admissions secondary to the studied condition or for other reasons?

Point 3 response: The manuscript has been edited to clarify the clinical data obtained: 

"A significant public health challenge is our ability to track the long term outcomes of these children regarding receipt of recommended care and development of complications.  On an individual basis, even this small demonstration project demonstrates the challenges of communication and care coordination for families and the medical home.  These public health and individual clinical barriers are not unique to these diagnoses, since the same systems of care exist for all children with conditions identified on NBS."

Please see the attachment for a copy of the clinical outcomes checklists for both SCD and DHH.   
